# Bioactive Compound Diversity in a Wide Panel of Sweet Potato (*Ipomoea batatas* L.) Cultivars: A Resource for Nutritional Food Development

**DOI:** 10.3390/metabo14100523

**Published:** 2024-09-26

**Authors:** Marion Nabot, Cyrielle Garcia, Marc Seguin, Julien Ricci, Catherine Brabet, Fabienne Remize

**Affiliations:** 1QualiSud, Univ Montpellier, CIRAD, Institut Agro, Avignon Université, Univ de La Réunion, F-34398 Montpellier, France; marion.nabot@univ-reunion.fr (M.N.); julien.ricci@cirad.fr (J.R.); catherine.brabet@cirad.fr (C.B.); 2CIRAD, UMR PVBMT, F-97410 Saint-Pierre, France; marc.seguin@cirad.fr; 3CIRAD, UMR QualiSud, F-97410 Saint-Pierre, France; 4SPO, Université Montpellier, Université de La Réunion, INRAE, Institut Agro Montpellier, F-34000 Montpellier, France; fabienne.remize@inrae.fr

**Keywords:** anthocyanins, antioxidant activity, carotenoids, fibers, phenolics, *Ipomoea batatas* L.

## Abstract

Objectives: This study provides an overview of the composition of the raw root flesh of a panel of 22 sweet potato (*Ipomoea batatas* L.) cultivars, with a focus on bioactive compounds. The large diversity of the proximate and phytochemical compositions observed between cultivars and within and between different flesh colors pointed out the importance of composition analysis and not only color choice for the design of foods with nutritional benefits. Methods: The nutritional composition (starch, protein, total dietary fibers) and bioactive compound composition of 22 cultivars from Reunion Island, maintained in the Vatel Biological Resource Center, were investigated. Results: Orange and purple cultivars stood out from white and yellow cultivars for their higher nutritional composition. Purple sweet potatoes were notable for their high contents of anthocyanins (55.7 to 143.4 mg/g dry weight (DW)) and phenolic compounds, in particular chlorogenic acid and ferulic acid, contributing to antioxidant activities, as well as their fiber content (14.1 ± 2.1% DW). Orange cultivars were rich in β-carotene (47.2 ± 0.7 mg/100 g DW) and to a lesser extent α-carotene (4.8 ± 1.2 mg/100 g DW). In contrast, certain white cultivars demonstrated suboptimal nutritional properties, rendering them less relevant even for applications where the lack of coloration in food is desired. Conclusions: Those characteristics enable the selection of sweet potato varieties to design food products ensuring optimal nutritional benefits and culinary versatility.

## 1. Introduction

Sweet potato (*Ipomoea batatas* L.) ranks among the most important food crops globally, following rice, wheat, potato, maize, and cassava [1]. In 2022, the annual production of sweet potatoes was estimated to exceed 86 million tons [2]. China is the largest sweet potato producer, contributing 54.2% of global production, followed by six African countries, which together account for 24.2% [2]. This crop plays a crucial role in ensuring global food security, particularly in Africa and other areas facing climate change impacts [3,4]. Sweet potato is characterized by its resilience to cultivation conditions, such as low soil fertility, drought, cyclones, heavy rainfall, or large temperature range, and by its ease of propagation. There are approximately 6500 sweet potato varieties worldwide, with root skin and flesh colors ranging from almost pure white through cream, yellow, orange, or pink to deep purple [5].

Sweet potato roots mainly contain starch, making up 50–80% of their dry weight (DW) [6]. Sweet potato starch presents a lower glycemic index (40 to 60 for boiled sweet potatoes) than potato starch (73 to 87 for boiled potatoes), which is beneficial for overweight and diabetic people [7]. In addition to starch, sweet potato roots contain other compounds of nutritional interest including dietary fibers, vitamins and phytochemicals such as carotenoids, anthocyanins, and phenolic acids. These phytochemicals are known for their antioxidant properties and associated health benefits [8]. Sweet potato flesh color is a crucial determinant of nutritional benefits [9]. While several white cultivars contain high starch levels, and purple cultivars are characterized by their high contents of antioxidant anthocyanins, the yellow and orange varieties provide carotenoids at levels that can help reduce vitamin A deficiencies.

The nutritional composition of sweet potato roots varies not only according to variety but also with environmental growing conditions, including climate and agricultural practices, and maturity [10]. Selecting plant cultivars with high nutritional value is essential for promoting a balanced diet conducive to optimal health, addressing a public health concern. This is particularly relevant in tropical and warm territories, where crops such as sweet potato are gaining popularity due to their versatility for both human and animal diets. In Reunion Island, located in a subtropical area, sweet potato roots are part of the traditional diet. The island’s culinary traditions are influenced by its colonization from Africa and Asia. As in other areas, sweet potato roots are typically consumed boiled, baked, fried, or steamed. Beyond the initial composition of the root, the cooking methods influence the retention of bioactive compounds [9]. The collection of the Vatel Biological Resource Center (Vatel BRC) maintains a diversity of cultivars [11]. However, there is limited information on the nutritional composition of these cultivars.

This study aimed to assess the diversity of the proximate composition, bioactive compounds, antioxidant activity, and starch gelatinization temperature in the roots of 22 sweet potato cultivars grown under the same environmental conditions. Leveraging the phenotypic characterization and nutritional composition of these cultivars would be relevant for selecting those best suited to design foods with enhanced nutritional benefits.

## 2. Materials and Methods

### 2.1. Sweet Potato Root Samples

Twenty-two sweet potato cultivars (four plants of each) from the BRC Vatel collection (Appendix A) were cultivated at the CIRAD experimental station located at Bassin plat, Saint Pierre in Reunion Island (21°19′21.8″ S 55°29′17.9″ E) for five months from March to August 2022. For each cultivar, at least 12 roots were harvested (minimum 3 roots/plant). Cultivation and harvest were performed at the same period to minimize the variability resulting from environmental conditions. The phenotypic traits of the leaves and roots were characterized visually using the list of sweet potato descriptors established by [12]: shape, color, and number of lobes for leaves; shape, skin, and flesh color for roots.

For each sweet potato cultivar, the roots were washed with tap water immediately after harvest, dried at room temperature, and cut in half lengthwise for the visual determination and colorimetric measurement of flesh color. Then the half-roots were peeled and cut into cubes with 1 cm sides. After manual homogenization, 60 g was used for the determination of DW, while the remaining root cubes were pooled and crushed in a grinder (IKA A11 basic, Staufen, Germany). For the determination of the starch, dietary fiber, and total phenolic contents as well as the quantification of phenolic compounds and the evaluation of free radical scavenging activity, 60 g of the crushed roots was kept at −80 °C for four hours, freeze dried using a Christ Alpha 1-2 LD plus (Osterode, Germany), and reduced to powder with a grinder (IKA A11 basic, Staufen, Germany). The rest of the crushed roots was stored in containers at −20 °C until the analysis of pH, titratable acidity, proteins, carotenoids, and anthocyanins. All the analyses were performed in triplicate on three samples of powdered or crushed and thawed roots.

### 2.2. Root Flesh Color Measurement

The L*a*b* parameters were determined with a Chroma Meter CR 400 (Konica Minolta Sensing Inc., Tokyo, Japan) applied on the surface of the sweet potato flesh, at three different points along the length of the root cut in half. For each cultivar, the color difference (DE) was calculated as follows using the white cultivar W1 as a reference:(1)$$\mathrm{DE}=\sum_{1}^{3}\;\surd (({L*}_{i}-{L*}_{0}{)}^{2}+({a*}_{i}-{a*}_{0}{)}^{2}+({b*}_{i}-{b*}_{0}{)}^{2})$$
in which _i_ corresponds to the cultivar and _0_ to the reference (mean of 3 measured points).

### 2.3. pH and Titratable Acidity

One gram of sweet potato crushed and thawed root sample was mixed with 10 mL of distilled water. The titratable acidity (TA) and pH were measured using a pH 2700 Eutech (EUTECH Instruments, Vernon Hills, IL, USA) equipped with an automatic titrator connected to a 0.05 N NaOH solution. TA was calculated as mg citric acid eq/100 g fresh weight (FW) according to the formula:TA = ((V_NaOH_ × C_NaOH_/3)/Pe) × Mw × 100(2)
in which V_NaOH_: volume of sodium hydroxide (mL); C_NaOH_: molar concentration of sodium hydroxide (0.05 mmol/mL); Pe: sample weight (g); and Mw: molecular weight of citric acid (192.12 g/mol).

### 2.4. Analysis of Proximate Composition

The DW was determined according to AOAC method 925.10 [13] with some modifications. Twenty grams of sweet potato root cubes was dried at 60 °C for 24 h and then at 105 °C for 48 h.

The protein content was determined according to the Bradford method [14] using one gram of crushed and thawed root sample.

The total starch content was determined following the two methods described by [15] for analyzing total carbohydrates and free glucose, using 500 mg and 100 mg of sweet potato root powder sample, respectively. The first method included two successive steps: the release of glucose by enzymes and a colorimetric reaction. Free glucose was determined using the enzyme–dye redox system. Glucose (0 to 37.5 µg/mL) was used as the standard. The total starch content, expressed as % DW, was calculated as total released glucose minus free glucose.

The total dietary fiber content was determined using the enzymatic–gravimetric AOAC method 991.43 [16] from one gram of sweet potato root powder sample.

### 2.5. Determination of Starch Gelatinization Temperature

The viscosity profile of a crushed and thawed root sample, without the addition of water, was obtained through a heating and cooling cycle using a Rapid Visco Analyzer (RVA) (Anton Paar MCR 301, Paris, France). Each sample (30 g) was kept at 50 °C for 2 min, then the temperature was increased to 95 °C in 22.5 min, and it was maintained at 95 °C for 1.5 min. Thereafter, the sample was cooled to 60 °C for 3 min. The starch gelatinization temperature was determined from the viscosity curve as the intersection point between the two lines tangent to viscosity = 0 (v0) and maximum viscosity (v_max_).

### 2.6. Carotenoid, Phenolic and Anthocyanin Total Contents

Total carotenoids were determined according to the method described by [17]. One gram of crushed and thawed root sample was homogenized with hexane/acetone/ethanol, (50:25:25, *v*/*v*/*v*) and sonicated (sonicator UP 200St, Hielscher, Imlab, Wasquehal, France) three times for 30 s at 160 W, 26 kHz. Then, the reaction mixture was centrifuged at 10,000× *g* for 10 min at 4 °C. The absorbance of the supernatant was measured at 450 nm. The total carotenoid content, expressed as mg β-carotene eq/100 g DW was calculated with a β-carotene molar absorption coefficient of 2505 expressed for 100 mL/g/cm in hexane [18].

Total phenolics were extracted from 50 mg of sweet potato root powder sample using the method described by Barral et al., 2019 [19], and quantified using the Folin–Ciocalteu method [20]. The total phenolic content, expressed as mg gallic acid eq (GAE)/100 g DW, was calculated using a calibration curve for gallic acid (0 to 50 µg/mL).

The total anthocyanin content (TAC) was determined according to the method described by [21], using one gram of crushed and thawed sample.

The TAC was expressed as mg eq cyanidin glucoside/g DW according to: TAC = (A × Mw × DF × 1000)/ε(3)
where DF: dilution factor; Mw: 449.2 g/mol for cyanidin glucoside; ε: 26.900 L/cm/mol for cyanidin glucoside.

### 2.7. Identification and Quantification of Phenolic Compounds

Phenolic compounds were identified and quantified using an ultimate 3000 HPLC (Dionex Co., Santa Clara, CA, USA) coupled to a PDA-3000 detector. A Waters Symmetry Shield reversed-phase C18 column 100 Å, 5 µm, 4.6 mm × 250 mm (Milford, CT, USA), was used and maintained at 30 °C. A volume of 20 µL of the previously extracted phenolics was injected by the autosampler. Mobile phase A consisted of formic acid and ultra-pure water 0.1%, and mobile phase B consisted of formic acid and methanol 0.1%. The gradient elution was adapted from [19] as follows: 5% B for 10 min, 7% B for 45 min, 35% B for 12 min, 80% B for 28 min, 100% B for 35 min, and 5% B for 10 min. The flow rate was 0.5 mL/min. Chromatographic data were recorded over the 240–320 nm range and integrated at 280 nm for phenolic compounds. The phenolic compounds were identified and quantified using standard solutions of gallic acid, chlorogenic acid, p-coumaric acid, *trans*-caffeic acid, and ferulic acid.

### 2.8. Identification and Quantification of Carotenoids

Carotenoids were identified and quantified using a UHPLC (LC-40, Shimadzu, Noisiel, France) coupled to an SPD-M40 and a YMC carotenoid C30 column (100 Å, 5 µm, 4.6 mm × 250 mm), maintained at 30 °C, according to [22]. A volume of 10 µL of the previous total carotenoids extract was injected by the autosampler. Mobile phase A consisted of methanol/methyl tertiary-butyl ether/water (96:2:2, *v*/*v*/*v*), and mobile phase B consisted of methyl tertiary-butyl ether/methanol/water (80:18:2, *v*/*v*/*v*). The gradient elution was as follows: 0% B for 1 min, 60% B for 9 min, 100% B for 20 min, and 0% B for 7 min. The flow rate was 0.5 mL/min. Chromatographic data were recorded over the 400–600 nm range and integrated at 450 nm for carotenoids. The latter were identified and quantified using standard solutions of β-carotene, lutein, zeaxanthin, and β-cryptoxanthin.

### 2.9. Free Radical Scavenging Activity

The free radical scavenging activity was evaluated using the DPPH (2.2-diphényl 1-pycrilhydrazyle) assay as described by [20] from 50 µL of the previous phenolics extract. The antioxidant activity, expressed as µg gallic acid eq (GAE)/g DW, was calculated using a gallic acid (0 to 50 µg/mL) standard curve.

### 2.10. Statistical Analysis

The results are expressed as mean ± standard deviation. Statistical differences between sweet potato cultivars were determined by an Analysis of Variance (ANOVA) followed by a Bonferroni test with a 95% confidence interval. Statistical differences between sweet potato flesh color groups were evaluated using the Kruskal–Wallis test followed by a Conover test. PCA (Principal Component Analysis) was used to determine correlations between variables and to group individuals by simplifying the number of dimensions. Statistical analyses were performed with R studio (version 4.3.0) statistical software with the packages ggplot, tidyverse, factomineR, and multicompview.

## 3. Results

### 3.1. Phenotypic Traits within the Cultivar Panel

The phenotypic traits of leaves and roots of the 22 sweet potato cultivars are presented in Appendix A and Appendix A. The shape and color of the leaves and root skin showed a large diversity, enabling the differentiation of most cultivars, but not W1 and W3.

The cultivars were categorized into four groups based on root flesh color, with significant differences in DE observed among these groups (white: 1.2–6.8, yellow: 5.8–23.7, orange: 49.8–55.8, and purple: 47.7–72.2) (Figure 1a). The L*, a*, and b* parameter values indicate the diversity of color properties among the sweet potato root cultivars (Appendix A and Table 1). White and yellow cultivars grouped together with the lightness (L*) and a* values, whereas orange and purple cultivars were in distinct groups (*p*-value < 0.01). The purple cultivar showed the widest range of L* and a* values, because of the presence of white veins in the flesh. The b* values were different between the four flesh color groups, being close to 0 (no yellow component) in purple cultivars and up to 54.9 for the orange cultivar O2.

The pH values of sweet potato roots ranged from 6.2 to 6.7 and did not differ between the flesh color groups (*p*-value = 0.18) (Appendix A). TA averages were 105.3 ± 26.2 mg citric acid eq/100 g FW, 115.8 ± 20.7 mg citric acid eq/100 g FW, 157.6 ± 12.2 mg citric acid eq/100 g FW, and 142.4 ± 47.0 mg citric acid eq/100 g FW for white, yellow, orange, and purple sweet potatoes, respectively (Figure 1b). The orange flesh group exhibited a higher TA than the white or yellow flesh groups.

### 3.2. Proximate Composition

The DW as well as total protein and starch contents in sweet potato roots varied between cultivars and within flesh color groups, but no significant differences were observed for these variables between flesh color groups (Appendix A). The DW ranged from 21.9 to 35% (variation coefficient, VC of 13%), and the total protein content varied from 2.1 to 9.2% DW, showing a more than 4-fold variation (VC 39%) (Figure 2a,b). The lowest protein contents were observed for cultivars W4, W7, and P1, with values below 2.5% DW, and the highest, above 8.0% DW, were observed for cultivars O2 and P2. Hence, the diversity in protein content between cultivars was noteworthy, though not related to flesh color. The starch content ranged from 52.0 to 83.4% DW, representing a nearly 2-fold difference among white sweet potato cultivars, with W5 having the lowest and W2 the highest contents.

The root fiber content ranged from 7.4% to 16.8% DW (Appendix A and Figure 2c). The purple and white sweet potato groups exhibited the extreme fiber contents of 14.1 ± 2.1% DW and 8.5 ± 0.9% DW, respectively. A variation coefficient between all cultivars of 25% was observed, and the cultivars with extreme fiber contents were W12 and P2.

### 3.3. Starch Gelatinization Temperature

The viscosity profiles as a function of temperature varied depending on the sweet potato cultivar (Figure 3a). For all, the viscosity remained stable at around 5000 mPa·s up to about 70 °C. Above this temperature, it increased to a maximum viscosity of approximately 25,000 mPa·s for the white varieties, 40,000 mPa·s for the orange, and 45,000 mPa·s for the yellow and purple varieties. As the temperature continued to rise, the viscosity decreased because of the rupture of swollen starch granules leading to gelatinization. No significant differences were observed for the starch gelatinization temperature between flesh color groups (Figure 3b). This temperature ranged from 70.4 to 85.8 °C and was between 74 and 77 °C for 17 of the 22 cultivars (Appendix A). The lowest and highest values were obtained for the white cultivar W11 and the orange cultivar O1, respectively.

### 3.4. Antioxidant Activity and Bioactive Compound Composition

The antioxidant activity ranged from 164.6 to 658.3 µg GAE/g DW, hence showing a 4-fold difference (Appendix A and Figure 4a). The yellow and purple flesh color groups exhibited higher antioxidant activity (average: 519.2 ± 110.9 µg GAE/g DW and 462.9 ± 51.8 µg GAE/g DW, respectively) than the white and orange flesh color groups (average: 360.4 ± 118.3 µg GAE/g DW and 349.7 ± 49.7 µg GAE/g DW, respectively). However, the cultivars with the highest antioxidant activity were P2, Y1, and Y2, and the ones with the lowest activities were W12 and W13.

Total carotenoids were found in variable concentrations among the different sweet potato cultivars (Appendix A and Figure 4b). As expected, differences were marked according to the flesh color. Orange sweet potatoes contained the highest levels of total carotenoids (from 56.9 to 62.2 mg β-carotene eq/100 g DW), whereas the values ranged from 1.0 to 3.7 mg β-carotene eq/100 g DW in yellow sweet potatoes and were below 1.2 mg β-carotene eq/100 g DW in white and purple sweet potatoes. Regarding the carotenoid composition, a major compound, β-carotene, and three minor carotenoids, α-carotene, lutein, and β-carotene 5,8 epoxide, were identified in orange and yellow flesh sweet potatoes (Table 1). Orange flesh cultivars were the richest in these carotenoids, with β-carotene (47.2 ± 0.7 mg β-carotene eq/100 g DW) representing 75 to 84% of the detected carotenoids. Yellow cultivars contained an average of 0.3 ± 0.2 mg β-carotene eq/100 g DW. The second most abundant carotenoid was α-carotene, with average values of 4.8 ± 1.2 mg β-carotene eq/100 g DW and 0.4 ± 0.5 mg β-carotene eq/100 g DW for orange and yellow sweet potatoes, respectively.

Anthocyanins were found in the purple cultivars at the highest contents (from 55.7 to 143.4 mg/g DW) and in one orange sweet potato cultivar (6.9 mg/g DW) (Appendix A, Figure 4c). A VC value of 35% was obtained for the anthocyanin content in purple cultivars, showing a more than 2-fold variation in this group. The highest value was recovered for P1.

The total phenolic content of the 22 sweet potato cultivars ranged from 153.5 to 416 mg GAE/100 g DW, with no significant differences observed between color groups (Figure 4d), but significant variations according to the cultivar were noted (Appendix A). The main diversity was observed within the yellow flesh cultivars, with a VC of 38%. This group included the two cultivars with the most extreme antioxidant activities, with Y1 having the highest and Y4 the lowest.

Five phenolic acids were identified and quantified in the cultivar flesh (Table 1): chlorogenic acid, ferulic acid, p-coumaric acid, *trans*-caffeic acid, and gallic acid. Chlorogenic acid was found mainly in the purple flesh cultivars. It was not detected in 11 cultivars of other color groups. Its highest content was in P2, with 949.6 ± 107.4 µg/g DW. Ferulic acid was quantified in all 22 cultivars and ranged from 34.1 to 327.3 µg/g DW. The purple flesh group, and especially the P2 cultivar, exhibited the highest ferulic acid concentrations, and the white ones exhibited the lowest. The compound p-coumaric acid was found in orange and in some purple cultivars, at a lower concentration in one yellow cultivar, and not in the white sweet potatoes. The highest p-coumaric acid contents were observed for cultivars P3 and O1, with more than 1024 µg/g DW, and P2, with 3285.0 ± 131.2 µg/g DW. *Trans*-caffeic acid was found in most white flesh cultivars and in some yellow and purple cultivars, but not in orange. The highest contents of *trans*-caffeic acid were determined in cultivars W4, W6, and P2 and were above 264 µg/g DW. Gallic acid was absent from purple flesh cultivars. Its content was higher in yellow and orange sweet potatoes (185 ± 178.2 µg/g DW and 83.7 ± 17 µg/g DW respectively) than in white cultivars (56.2 ± 20.7 µg/g DW). Cultivars Y3 and Y4 contained more than 101 µg/g DW of gallic acid.

White, yellow, and orange cultivars contained mainly ferulic and gallic acids. In contrast, high levels of chlorogenic and ferulic acids and the absence of gallic acid characterized the purple cultivars. More specifically, the P2 cultivar, and to a minor extent the P3 one, contained high levels of p-coumaric and *trans*-caffeic acids.

### 3.5. Principal Component Analysis

The values of the determined characteristics were used as variables for the 22 sweet potato cultivars that were subjected to PCA. Three principal components (PC) out of the 22 had eigenvalues greater than 1. Two PCs, explaining approximately 57% of the total variance, were selected for visual 2D representation (Figure 5). The correlation coefficients between the variables are displayed in Appendix A.

As expected, four groups stood out, each corresponding to a sweet potato flesh color group. The variables contributing most to the formation of group 1 (orange sweet potato cultivars) were the contents of total carotenoids and carotenoid compounds (β-carotene, α-carotene, lutein, and β-carotene 5,8 epoxide). Pearson correlation tests (rho > 0.6; *p*-value < 0.05) revealed that carotenoid compounds evolved concomitantly and were positively correlated with the total carotenoid content and color difference. Group 2 (purple sweet potato cultivars) was characterized by high fiber, anthocyanin, and three phenolic acid (chlorogenic, ferulic, and p-coumaric acids) contents and antioxidant activity. These three phenolic acids exhibited positive correlations with anthocyanins but also with the purple sweet potato color. Overall, the color difference compared to the white cultivar W1 was positively correlated with the fiber content and could serve as an indicator of fiber richness.

Lastly, the gallic acid, total phenolic, and starch contents as well as the pH value clustered closely together in the PCA representation, where the white and yellow cultivars were grouped. Total phenolics were negatively correlated with DW.

## 4. Discussion

This study explored the physicochemical properties and nutritional composition of 22 sweet potato cultivars from Reunion Island, revealing significant diversity despite their cultivation under the same environmental conditions. Our study provided robust data on the composition diversity within and across the flesh color of cultivars. The analysis showed distinct differences in nutrients and bioactive compounds across flesh color groups, particularly in terms of total anthocyanins, carotenoids, fiber content, antioxidant capacity, and specific phenolic acids. The physicochemical characteristics, macronutrient content, and gelatinization temperatures were consistent with previous studies [8,23,24,25,26].

A recent review underlined the significant variation in the phenolic compound profiles between purple, orange, yellow, and white sweet potatoes, highlighting the unique nutritional benefits of each variety [27]. However, marked differences were observed with previous studies, as ferulic and gallic acids were found at high levels in yellow and orange cultivars in our study, whereas caffeic, chlorogenic, and caffeoylquinic acid derivatives were reported for these sweet potato colors in [27]. Chlorogenic acid, ferulic acid, and p-coumaric acid found at high levels in purple flesh are formed by the condensation of quinic acid and *trans*-cinnamic acids. In the literature, 3-O-caffeoylquinic acid (3-CQA) and 5-CQA are often prevalent in purple sweet potatoes [8], but they were not identified in this study. The predominance of chlorogenic, ferulic, and p-coumaric acids in purple sweet potatoes is due to the stability afforded by their interaction with anthocyanins [27]. We observed that the color differences were correlated with the total anthocyanin content, ferulic acid, chlorogenic acid, and p-coumaric acid contents but not with the *trans*-caffeic acid content (Appendix A). These bioactive compounds are not only essential for enhancing the nutritional value of sweet potatoes but also offer significant health benefits, such as antioxidant protection and anti-inflammatory effects. This diversity highlights the potential of these cultivars for developing nutrient-rich food products with targeted health benefits.

Orange cultivars stood out due to their high carotenoid content, particularly β-carotene, making them excellent candidates for addressing vitamin A deficiencies [10,28]. As previously reported, the use of orange flesh varieties is particularly potent to improve food and nutritional security [9]. In our study, a pro-vitamin A activity of approximately 8.3 retinol equivalents was calculated for the orange cultivars. Previously, a wide range of total carotenoids (0.85 mg/100 g FW to 6.71 mg/100 g FW) was reported in orange cultivars [10,29]. Those values correspond to the one we reported for yellow cultivars, with the analyzed orange ones being far greater.

Purple cultivars, on the other hand, were rich in anthocyanins and phenolic acids, which contribute to their strong antioxidant activity. The bright hues of the purple flesh are attributed to anthocyanin pigments. The anthocyanin properties vary according to endogenous factors such as pH and co-pigmentation, resulting in different colors and shades (red, blue, and colorless to slight yellow) despite close concentrations. Anthocyanins are seen as mono-, di-, and non-acylated forms with peonidin, cyanidin, or pelargonidin aglycones [30]. In [8,30], the anthocyanin content in purple cultivars varied from 8.5 to 13.9 mg/g DW. These values are 4- to 10-fold lower than those found in the present study. Phenolic acids and diacylated anthocyanins are involved in the high antioxidant activity [8]. Finally, the purple sweet potatoes analyzed in this study contained a high fiber content, aligning with the previous literature (2.1–13.6% DW) [24,31]. The fibers, classified as non-digestible carbohydrates, offer various health benefits, including reducing intestinal transit time, managing body weight, modulating the gut microbiota, and reducing cholesterol reabsorption. Both insoluble compounds such as lignin, cellulose, and hemicellulose and soluble compounds such as pectin contribute to the dietary fiber fraction in sweet potatoes, with variations in the composition across different cultivars [32]. Within the purple group, the P2 cultivar is an outsider with numerous advantages: high protein content, high fiber content, high antioxidant activity, and highest contents in all phenolic acids detected in the group. This finding highlights the potential of these purple cultivars as functional foods because of the physiological benefits they can provide.

This study also found that white and yellow cultivars, while nutritionally less advantageous, could still contribute to a diverse diet, albeit with lower concentrations of carotenoid and anthocyanin bioactive compounds. As previously reported, white cultivars provide high starch levels with a low glycemic index [3,9]. However, within those groups, several cultivars, especially W12, are less relevant for use as a high-nutritional-quality food because of the low fiber or protein contents or the low antioxidant activity.

The variation in nutritional profiles across these cultivars underscores the importance of selecting sweet potato varieties based on their specific bioactive compound content to maximize nutritional benefits. Correlation analysis showed that color is a crucial indicator for major bioactive compounds such as anthocyanins and carotenoids or fibers but fails to accurately identify cultivars’ nutritional properties, especially within pale flesh cultivars. In addition, although sweet potato roots can sometimes be consumed raw, cooking impacts their composition [4]. Careful evaluation of the transformation method is necessary to preserve the benefits of the nutritional and phytochemical properties. In that view, lactic acid fermentation is considered a promising way to retain or even increase the health benefits of sweet potatoes [33].

Attributes such as bright colors, appealing textures, and balanced acidity levels are sought in various food products to enhance their sensory appeal and consumer acceptance. Acidity levels in sweet potato roots play a marked role in food acceptability and product quality, influencing their palatability and perceived freshness [34,35]. The acidity levels observed in white and yellow sweet potatoes were consistent with these previous studies. Concerning the texture-related aspects, starch, as a major component in sweet potatoes, is a key-factor of their functional properties. Its molecular composition, particularly the amylose/amylopectin ratio, is crucial for processing suitability, influencing starch rheological properties, gelatinization, and retrogradation profile [36]. Studies have reported variable amylose/amylopectin ratios between different sweet potato flesh color groups [37] and within the same color group [38]. The starch gelatinization temperature, indicating the transition from a crystalline to a gelatinized state, is an important parameter in starch characterization [6]. This transition significantly affects starch’s functional properties, such as the viscosity, stability, and water-holding capacity. Moreover, the gelatinization temperature offers insights into the starch composition, structure, and thermal decomposition behavior [6]. In the present study, an original methodology was applied to crushed sweet potatoes instead of flour mixed with water. The findings align with those in the literature [37,39], indicating a consistent range of starch gelatinization temperatures for the sweet potato cultivars. Thus, this study highlighted the influence of starch composition on the functional properties of sweet potatoes, particularly in terms of processing suitability and textural qualities. This is critical for the development of food products that are not only nutritionally beneficial but also meet consumer preferences for taste and texture.

The classification of sweet potato cultivars based solely on phenotypic features such as leaf or root shape and color are often insufficient to distinguish sweet potato cultivars. This was shown in the case of cultivars W1 and W3, which had the same morphotype but differed significantly in protein, starch, and phenolic compound (*trans*-caffeic acid) contents. These differences likely arise from genomic polymorphism, indicating the need for genetic analyses to elucidate cultivar diversity and guide nutritional assessments and functional applications. This need becomes even more pronounced given the variability in secondary metabolite composition and antioxidant activity between cultivars, influenced by factors such as growing environment and harvest maturity. Consequently, accurate discrimination of cultivars through genetic analysis is helpful to assess their potential for food applications.

Overall, this research demonstrates the value of integrating phenotypic and nutritional data to guide the selection of sweet potato cultivars for food applications. The diversity in bioactive compounds across and within these cultivars offers a rich resource for developing foods with enhanced nutritional benefits. Genetic analyses, alongside phenotypic characterization, are recommended to further refine the selection process and fully harness the potential of sweet potato diversity in addressing nutritional needs and promoting health.

## 5. Conclusions

Integrating the phenotypic characteristics and nutritional composition of sweet potatoes is crucial for effectively selecting cultivars that are best suited for developing nutrient-rich food products. By leveraging both the color properties and bioactive compound profiles, farmers can make informed choices that align with consumer demand for diverse, health-enhancing food options. In particular, the knowledge of the composition of local varieties is a valuable tool for promoting the cultivation of beneficial cultivars among farmers. It can also provide food stakeholders and consumers with important information to support a better diet, both for sustainability and health-related considerations. This comprehensive approach not only elevates food quality but also maximizes the nutritional benefits that sweet potatoes can offer, thereby supporting overall health and well-being. In this study, purple cultivars are particularly recommended for diversifying processed products due to their superior nutritional value and higher concentrations of bioactive compounds compared to white and yellow cultivars.

## Figures and Tables

**Figure 1 metabolites-14-00523-f001:**
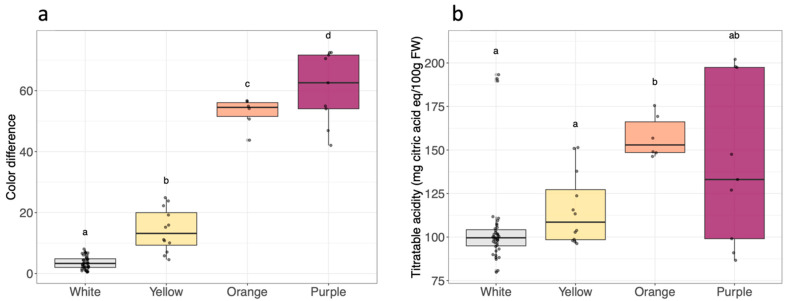
(**a**) Color difference, (**b**) titratable acidity according to sweet potato root flesh color (white *n* = 13, yellow *n* = 4, orange *n* = 2, purple *n* = 3). In the boxplots, the black line inside the box indicates the median, and the bold black dot inside the box indicates the mean. The dots along the boxplot represent the values. The vertical lines outside the box represent the standard deviation of the mean. Distinct letters correspond to significantly different values analyzed by the Kruskal–Wallis test (*p*-value < 0.01).

**Figure 2 metabolites-14-00523-f002:**
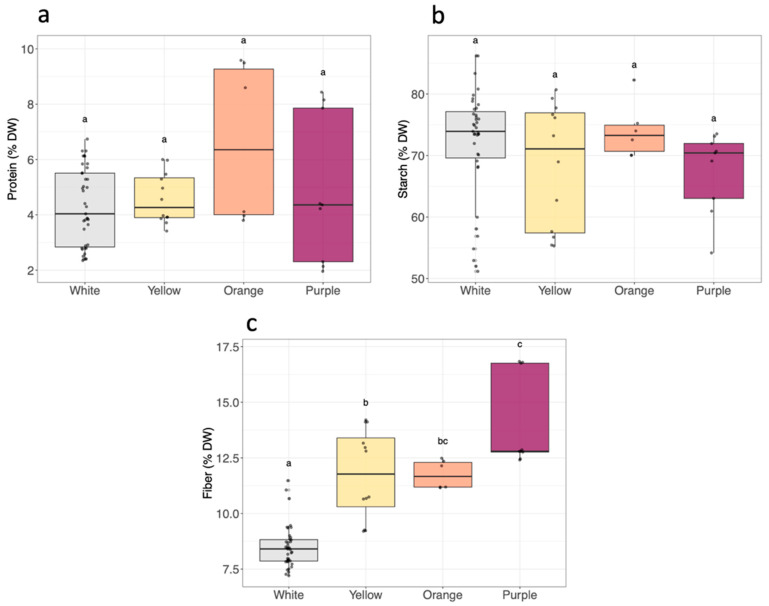
(**a**) Protein, (**b**) starch, and (**c**) fiber according to sweet potato root flesh color (white *n* = 13, yellow *n* = 4, orange *n* = 2, purple *n* = 3). In the boxplots, the black line inside the box indicates the median, and the bold black dot inside the box indicates the mean. The dots along the boxplot represent the values. The vertical lines outside the box represent the standard deviation of the mean. Distinct letters correspond to significantly different values analyzed by the Kruskal–Wallis tests (*p*-value < 0.05).

**Figure 3 metabolites-14-00523-f003:**
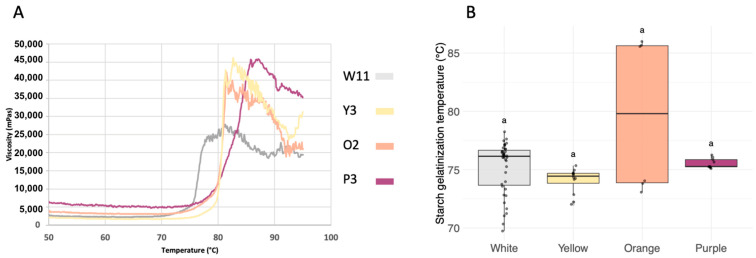
(**a**) Typical viscosity profiles and (**b**) starch gelatinization temperature according to sweet potato root flesh color (white *n* = 13, yellow *n* = 4, orange *n* = 2, purple *n* = 3). In the boxplots, the black line inside the box indicates the median, and the bold black dot inside the box indicates the mean. The dots along the boxplot represent the values. The vertical lines outside the box represent the standard deviation of the mean. Distinct letters corresponded to significantly different values analyzed by the Kruskal–Wallis test (*p*-value < 0.05).

**Figure 4 metabolites-14-00523-f004:**
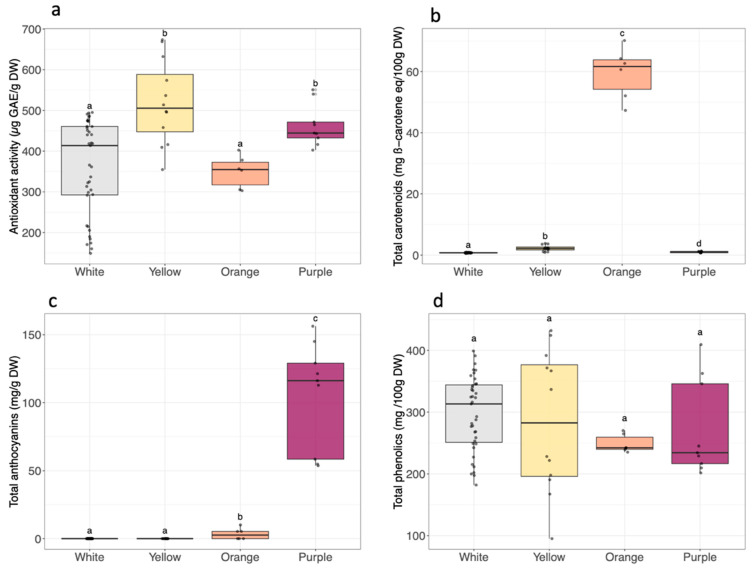
(**a**) Antioxidant activity, (**b**) total carotenoids, (**c**) total anthocyanins, and (**d**) total phenolics according to sweet potato root flesh color (white *n* = 13, yellow *n* = 4, orange *n* = 2, purple *n* = 3). Total phenolics are expressed as mg GAE/100 g DW. In the boxplots, the black line inside the box indicates the median, and the bold black dot inside the box indicates the mean. The dots along the boxplot represent the values. The vertical lines outside the box represent the standard deviation of the mean. Distinct letters correspond to significantly different values analyzed by the Kruskal–Wallis test (*p*-value < 0.05).

**Figure 5 metabolites-14-00523-f005:**
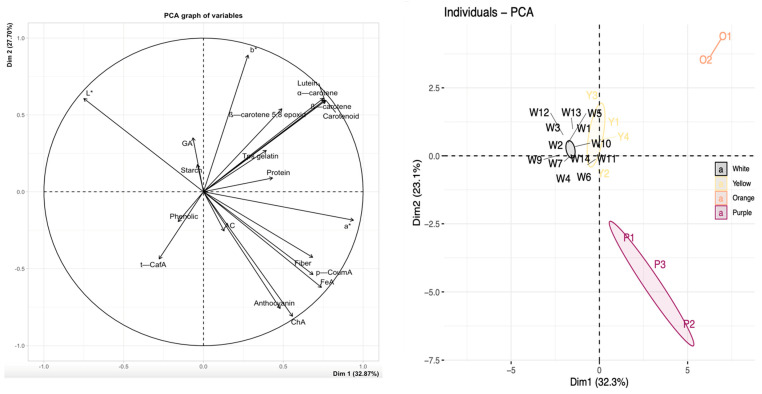
Principal component analysis (PCA) of the relative difference in the composition of the analyzed variables for the 22 sweet potato cultivars. TA: titratable acidity, AC: antioxidant capacity, ChA: chlorogenic acid, FeA: ferulic acid, GA: gallic acid, p-CoumA: p-coumaric acid, t-CafA: *trans*-caffeic acid, T° gelatin: starch gelatinization temperature.

**Table 1 metabolites-14-00523-t001:** Color properties (variation coefficient) and phytochemical composition of sweet potato root flesh according to cultivar color groups.

Sweet Potato Flesh Color	All	White	Yellow	Orange	Purple
L*	78.4 (22%)	86.9 ^A^ (2%)	85.2 ^A^ (3%)	67.3 ^B^ (2%)	39.5 ^C^ (26%)
a*	4.8 (285%)	−2.6 ^A^ (41%)	−2.3 ^A^ (36%)	32.5 ^B^ (8%)	27.9 ^C^ (21%)
b*	20.2 (72%)	16.8 ^B^ (18%)	32.2 ^C^ (22%)	51.5 ^D^ (11%)	−2.3 ^A^ (231%)
β-carotene (mg/100 g DW)	4.3 (316%)	ND	0.3 ± 0.2 ^B^	47.2 ± 0.7 ^A^	ND
α-carotene (mg/100 g DW)	-	ND	0.4 ± 0.5 ^B^	4.8 ± 1.2 ^A^	ND
β-car. 5,8 epoxide (mg/100 g DW)	-	ND	0.08 ± 0.07 ^B^	0.1 ± 0.1 ^A^	ND
lutein (mg/100 g DW)	-	ND	0.03 ± 0.02 ^B^	1.3 ± 0.4 ^A^	ND
chlorogenic acid (µg/g DW)	122.6 (207%)	24.6 ± 33.6 ^A^	7.7 ± 13.9 ^A^	81.1 ± 89.3 ^AB^	728.3 ± 175.1 ^C^
ferulic acid (µg/g DW)	106.6 (74%)	60.1 ± 18.1 ^A^	106 ± 26.5 ^B^	148.2 ± 21.1 ^C^	281.1 ± 34.9 ^D^
p-coumaric acid (µg/g DW)	118.7 (84%)	ND	89.2 ± 162.5 ^AB^	805.2 ± 246.2 ^B^	1606.4 ± 1425.3 ^B^
*trans*-caffeic acid (µg/g DW)	308.5 (247%)	147.1 ± 92.9 ^A^	74.8 ± 87.2 ^AB^	ND	133.3 ± 146.1 ^AB^
gallic acid (µg/g DW)	74.4 (119%)	56.2 ± 20.7 ^A^	185 ± 178.2 ^B^	83.7 ±17 ^B^	ND

^A,B,C,D^: Distinct upper-case letters correspond to significantly different values between color groups analyzed by the Kruskal–Wallis test (*p*-value < 0.01). DW: dry weight, ND: Not detected.

## Data Availability

The original contributions presented in the study are included in the article or Appendix A, further inquiries can be directed to the corresponding author.

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
