# Peer review of "Bioactive Compound Diversity in a Wide Panel of Sweet Potato (Ipomoea batatas L.) Cultivars: A Resource for Nutritional Food Development"

_metabolites, 2024, doi:10.3390/metabo14100523_

Round 1
Reviewer 1 Report
Comments and Suggestions for Authors
The selection of plant varieties for cultivation and breeding is a pressing issue of our time, aimed at the rational use of soils and other resources. A number of questions arose regarding the authors' article that require clarification:
1. In the work, starch is determined after freezing the plant material. As it known, low temperatures lead to a change in the structure of starch or its partial destruction. Does the results obtained in the work take this factor into account, were the experiments carried out without freezing?
2. The fact of fruit copigmentation caused by the interaction of anthocyanins and phenolic compounds, in particular flavonoids and phenolic acids (https://doi.org/10.3390/antiox10121967) is known. Was there a correlation between the content of measured phenolic acids, TAC and the color of the tubers?
3. Pay attention to the description of the HPLC gradient elution mode, is the method really that long (more than 2 hours)?
4. What is the reason for choosing standards for determining phenolic compounds in samples? The study uses only phenolic acids, compounds of other classes are not represented. It would be more informative to use mass spectrometry to identify phenolic compounds.
5. Check the flow rate in the experiment to determine phenolic compounds, is the specified flow rate of 0.3 ml/min optimal for a 5 μm column? When using this type of column, a flow rate of about 1 ml/min is more often used
Reviewer 2 Report
Comments and Suggestions for Authors
The paper is a nice addition to the literature concerning the properties of food ingredients, especially those that are crucial in the Global South.
The study is well structured, the methodology sound and the findings relevant.
I would suggest the authors make a few minor improvements:
0. I would add in the title and abstract the Latin binomial of the species
1. overall, the article would shine more if the authors could add a few explicative figures of the considered varieties and of their traditional preparations/dishes
2. In the introduction, more is to be said, in my humble opinion, about a. the phytochemistry of the genus b. the uses of this sp. in the Global South and esp. about its importance in ensuring food sucrtity
3. In the methods, authors should clarify more convincingly why they picked the specific assays they adopted, not others.
4. In the discussion, the link to food security and sovereignty has to be made more stringent, especially considering the findings that non-white tubers show a remarkable amount of bioactive compounds; the authors should also better illustrate the meaning of their main findings for nutritionists and food chemists in their future trajectories of research.
5. I would also suggest the authors to consider to shift a bit their discussion, i.e. why these studies are important and which are the concrete implications of these for traditional healthy and sustainable diets, local food systems, and the alleviation of malnutrition; in other words, I would advocated for a much clearer (even if short) message dovoted to local food stakeholders and policy decision makers in the agricultural domain.
Comments on the Quality of English Language
The English is very appropaite and would need only a minor editing
Round 2
Reviewer 1 Report
Comments and Suggestions for Authors
The authors took into account all the comments and made the necessary corrections